# How Can Personality Enhance Sustainable Career Management? The Mediation Effects of Future Time Perspective in Career Decisions

**In-Jo Park \*, Meiqiao Gu \* and Shenyang Hai**

Department of Psychology, Henan University, 1 Jinming St., Kaifeng 475004, China; hai1340057937@konkuk.ac.kr
\*  Correspondence: park@henu.edu.cn (I.-J.P.); 10030019@henu.edu.cn (M.G.); Tel.: +86-130-6931-7408 (I.-J.P.)

**Abstract:** This study seeks to explore the mediating effects of future time perspective (FTP) between personality variables and career decision-making self-efficacy and career indecision with respect to managing sustainable careers. We used an online survey to collect data from 250 undergraduates for Study 1, in which we explored the mediating role of FTP which focused on the emotional and cognitive personality traits of emotional intelligence, ego resilience, and self-control; and from 249 undergraduate students for Study 2, in which we investigated the mediating effects of FTP on the personality traits of extraversion, conscientiousness, and neuroticism. The results from the first study indicated that emotional intelligence, ego resilience, and self-control had indirect effects on career decision-making self-efficacy and career indecision via FTP. The results of the second study showed that extraversion, conscientiousness, and neuroticism had indirect effects on career decision self-efficacy and career indecision via FTP. These results contribute to an enhanced understanding of the relationship between personalities and career decisions, and they expand our knowledge about the antecedents and consequences of FTP. At the end of this paper, we discuss the theoretical and practical implications of this study and identify directions for future research.

**Keywords:**  future time perspective; personality; career decision-making; self-efficacy; career indecision

---

## 1. Introduction

University students intending to embark on sustainable, coherent career paths rather than just a series of jobs accepted for situational reasons (e.g., convenience to home) have the greatest chance of success if they decide on their prospective occupation before or during their college years. Yet evidence has shown that increasing numbers of undergraduates do not engage in career decision-making or have difficulty making career decisions until graduation [1], which can increase employment risks and psychological burdens [2]. Since early career decisions are perceived to be crucial to the long-term career success undergraduates [2], good career decisions can contribute to sustainable career development. Future time perspective (FTP) is considered as an important factor that influences individuals' career decisions and long-term development [3]. That is, FTP is predominantly conducted in career decision-making process [4–6]. Individuals with high positive expectations for future events choose to set higher goals and will achieve them excellently [7]. Although the time of career decision-making is essential, some undergraduates lacking confidence still delay, potentially limiting their career development. The planning attitude makes us focus on the future. That is, if individuals with this attitude are future oriented, they are more willing to prepare for their future as they possess a clear understanding of how their current activities affect the realization of their future goals [5]. Therefore, FTP could be a crucial factor in determining the effectiveness of career decisions among undergraduates [2,6].

In recent years, increasing research has focused on the relationship between personality factors and career decisions, demonstrating the important role of personality factors in facilitating individuals' career decisions [8–11]. However, little is known about why personalities predict career decisions. We expect that FTP may play an important role in the relationship between personality factors and career decision-making outcomes. That is, we aim to uncover the mechanisms why personality factors influence career decisions, as assuming FTP as a mediator. FTP refers to an individual's belief about the time left in the future and how an individual think about the time left [12]. Individuals with high FTP are likely to perceive future time as expansive, to highly value setting goals, and to actively engage in current activities to achieve their goals [13–15]. Zimbardo and Boyd (2015) [16] linked time perspective to personality variables such as openness, aggression, depression, self-esteem. Previous studies also found that FTP is positively associated with career decision outcomes, including career decision-making self-efficacy, career indecision, career decision-making difficulties, and career choice anxiety [3,6,17]. Accordingly, we propose that individuals' personalities would influence how they view their future time, which in turn influences their career decisions.

In the present study, we aim to examine the mediating effects of FTP in the relationships between emotional and cognitive personality, trait personality, and career decision-making, and provide a more systematic and deeper understanding of the role of FTP in the career decision-making process. This study focuses on the characteristics of career decision-making, including career decision-making self-efficacy and career indecision. Career decision-making self-efficacy is defined as the confidence or belief in one's career decision-making ability [18]. Career indecision is defined as one's inability to make an effective career decision [19]. It is generally believed that career decision-making self-efficacy is related to career indecision [20]. Thus, this study investigates the characteristics of the career decision-making, not only including the career decision-making self-efficacy, but also career indecision, focusing on both positive and negative aspects of career decision-making.

The purpose of this study is to investigate the mediating role of FTP between personality variables and career decision-making. Our study seeks to contribute to the literature on career decisions by providing a deeper understanding of the role of personality factors and FTP in the process of career decision-making. Moreover, considering the potentially positive and cognitive-motivational effects of FTP on an individuals' future development [21], we aim to further our knowledge of the antecedents and consequences of FTP, and provide implications for the sustainable career management. We designed two studies: study 1, in which we explore the mediating effects of FTP in the association between emotional personality factor (i.e., emotional intelligence), cognitive personality factors (i.e., ego-resilience, and self-control), and career decision outcomes, including career decision-making self-efficacy and career indecision, and study 2, in which we test the mediating role of FTP between personality traits (i.e., extraversion, neuroticism, and conscientiousness) and career decision-making.

## 2. Theoretical Background and Hypotheses

### 2.1. The Effects of FTP

Time perspective theory has attracted growing attention from researchers over the past decade of personality psychology, educational psychology, and organizational psychology. Some studies have focused on motivations, healthcare, and career development [13,22,23]. Zimbardo, Keough, and Boyd (1997) [24] divided time perspective into the past, present, and future, stating that individuals' time perspective influences their actions and how they adapt to change. Zimbardo and Boyd (2015) [16] (p. 1271) defined time perspective as the "often nonconscious process whereby the continual flows of personal and social experiences are assigned to temporal categories, or timeframes, that help to give order, coherence, and meaning to those events", making time perspective a basic aspect of people's subjective experiences. Past-oriented individuals tend to focus on traditions, family, and history; future-oriented individuals tend to focus on planning, goal setting, and goal attainment [4,25]. Some

research has shown that FTP is positively associated with academic and career-related success [26]; therefore, this study focused mainly on seeking links between FTP and sustainable career management.

FTP has two dimensions on which is focuses: opportunity and the time remaining [12]. According to Park (2014) [27], FTP constitutes individuals' perspective of their chances for future career success and an evaluation of their prospective careers, as well as the connection between their current activities and future careers. For the purposes of our study, we defined FTP as one's belief in future goals and the amount of time one considers he or she has left in the future, focusing on both opportunity and time remaining—focusing on both opportunity and time remaining. Studies suggest that individuals with higher FTP tend not only to view their future time as expansive, but also attach more value to their goals [5,21]. Previous studies have also found that FTP positively influences learning, mature experiences, and school life [28–30]. For example, Peetsma and Van der Veen (2011) [31] showed that having a positive FTP positively affected learning ability; the more students were oriented toward the future, the more diligently they studied, which led to better grades.

De Bilde, Vansteenkiste, and Lens (2011) [32], examining that FTP and cognitive information processing, found that individuals with a more positive FTP more effectively used cognitive information and more actively engaged in current activities to achieve their goals. Lang and Carstensen (2002) [33] found that individuals who thought of their future time as limited prioritized emotionally meaningful goals, and individuals who thought of their futures as open-ended prioritized instrumental or knowledge-related goals. Moreover, Tabachnick, Miller, and Relyea (2008) [34] revealed the motivating effects of FTP: individuals with a positive FTP attached more value to their current work and goals than those with a limited FTP, which increased their motivation. Miller and Brickman (2004) [35] and others [36] found that the anticipation of future outcomes is important in self-regulation; by setting proximal goals, individuals can link their current efforts to the attainment of valuable distal goals. Accordingly, we expected a positive FTP to have a positive relationship with scheduling future career decisions and regulating goal-oriented behavior.

## 2.2. FTP and Emotional and Cognitive Personality Traits

The current study took emotional intelligence as the variable for emotional personality factors, and ego resilience and self-control as the variables for cognitive personality factors. Ego resilience and self-control are cognitive personality factors, while emotional intelligence is an emotional personality factor [37]. Emotional intelligence is defined as individuals' ability to effectively recognize, express, understand, and regulate their own emotions [38,39]. Individuals with higher levels of emotional intelligence are more aware of their emotions, better able to integrate emotional experiences with thoughts and actions, and more likely to receive social support [40,41]. Emotional intelligence can help individuals cope, which gives the perception of social support [42]. Accordingly, individuals with high emotional intelligence are more likely to perceive the environment as supportive and see better future prospects.

The understanding and regulations of emotions are expected to influence individuals' perceptions of the future. Stolarski, Bitner, and Zimbardo (2011) [43] suggested that the capacity to understand and control emotions plays an important role in ensuring a reasonable balance between present pleasures and future consequences. Thus, individuals with high emotional intelligence are more likely to manage their emotions effectively and have a positive attitude toward the past, present, and future [43]. They may also be more adept at setting motivational goals, aims, and missions [44]. Mayer et al. (2004) [44] indicated that people who can understand and regulate their own emotions will consider the future in a more leisurely way and develop more plans, leading to increased future opportunities. By expressing and adjusting emotions effectively, individuals are more likely to have a positive attitude and evaluation of the value of the future. Accordingly, we expected that emotional intelligence would be positively related to FTP.

## 2.3. FTP and Career Decision

FTP is an important factor that influences career decision-making [3,6]. Park et al. (2018) [5] indicated that individuals with a positive FTP are more motivated to develop their careers and more confident in their ability to complete specific tasks. Moreover, individuals who are future-oriented tend to actively set expectations and goals, control their behaviors, and continually evaluate performance in their work [2,22]. In contrast, individuals who do not think in terms of time have difficulties getting over their current problems [45,46]. Accordingly, positive FTP is likely to motivate individuals to exert more effort toward overcoming obstacles and achieving goals, and this may increase their confidence in career decision-making and decrease their level of career indecision.

Walker and Tracey (2012) [6] linked FTP to career decision-making based on social-cognitive career theory [47–49]. They argued that the basic element of self-efficacy is perceiving the future as valuable, and this is closely relevant to future achievements: individuals who lack an understanding of how their current behaviors affect their goal achievement will not try as hard to complete current tasks, lowering their self-efficacy and making them less likely to complete their tasks effectively. Jung et al. (2015) [2] and Walker and Tracey (2012) [6] both demonstrated this significant relationship between FTP and career decision-making self-efficacy. Using data from first-year university students, Savickas et al. (1984) [46] also found that time perspective was negatively associated with career indecision. Ban (2014) [50] likewise demonstrated this negative relationship between FTP and career indecision among Korean college students. These findings suggest that individuals with positive FTP would display higher career decision-making self-efficacy and lower career indecision. Therefore, we developed these hypothesizes:

**Hypothesis 1a.** *FTP will mediate the relationship between emotional intelligence and career decision-making self-efficacy.*

**Hypothesis 1b.** *FTP will mediate the relationship between emotional intelligence and career indecision.*

Ego resilience was chosen as an antecedent of FTP for this study because ego resilience is highly related to positive attitude toward the future (e.g., optimism) and the ability to adapt [51,52], which may influence FTP. Ego resilience describes individuals' capacity to modify their modal level of self-control in response to situational needs and affordances [53,54]. Individuals with high ego resilience can adapt at changing circumstances and use problem-solving strategies flexibly; individuals with low ego resilience have less adaptive flexibility to deal with changing circumstances effectively [51,53]. Baldwin, Jackson, Okoh and Cannon (2011) [55] reported that the psychological resources of ego resiliency and optimism could reduce psychological distress. Moreover, ego resilience has been found to be positively associated with global adjustments, work and social adjustments, and psychological and physical health adjustments [52].

Watson and Tellegen (1985) [56] found that individuals with high ego resilience experienced more positive affect and had an absence of anxiety. Further, they have been reported as being more optimistic, self-disciplined, and self-motivated [52]. Optimism as a positive attitude is related to having a future-oriented perspective [57]. Optimistic people expect things to go well, whereas pessimistic people expect things to go badly [58]. Since individuals with high ego resilience are more optimistic and have higher adaptive flexibility [51,52], we expected that ego resilience would influence career decision-making self-efficacy and career indecision via FTP. Therefore, we added the following hypotheses:

**Hypothesis 1c.** *FTP will mediate the relationship between ego resilience and career decision-making self-efficacy.*

**Hypothesis 1d.** *FTP will mediate the relationship between ego resilience and career indecision.*

We selected self-control as a predictor of FTP since self-control is related to the regulation of cognition and behavior for goal attainments [54,59], which in turn, may influence people's perception of the future. Self-control refers to the ability to regulate emotions, dominate cognitions, inhibit impulses, and adjust behaviors to achieve long-term goals [54,60]. Previous research has linked self-control to positive outcomes, such as stable emotional experience, good adjustment, interpersonal success, and academic success [61,62]. Individuals with high self-control think what they do now is related to their future, identify the conflict between temptation and better judgment, and ably implement self-control strategies to achieve their goals. Accordingly, we expected that individuals with high self-control would be future oriented and have clear plans.

Research has indicated that individuals with high self-control employ better strategies that facilitate goal progress and accomplishment [59]. Furthermore, Eisenberg et al. (2009) [51] demonstrated that self-control could help individuals develop and improve their own skills, including attention control and planning, which may be positively associated with FTP. The capacity of individuals to shift their attention toward positive thoughts and events may make them more optimistic about the future, and this has been related to decreased anxiety and depression [63,64]. Planning is considered to be the preferred behavior or strategy of future-oriented individuals to achieve goals [2]. Therefore, individuals with high self-control effectively regulate their behaviors and attention, and engage in more planning to achieve their goals, and this is positively related to FTP. As a result, we proposed the following additional hypotheses:

**Hypothesis 1e.** *FTP will mediate the relationship between self-control and career decision-making self-efficacy.*

**Hypothesis 1f.** *FTP will mediate the relationship between self-control and career indecision.*

*2.4. FTP and Personality Traits*

In this study, we investigated the relationship between FTP and the personality traits extraversion, conscientiousness, and neuroticism. Extraversion is described as being active, energetic, gregarious, and assertive, as well as the tendency to experience positive affect [65,66]. Extroverts are self-confident, sociable, and proactive, so that they tend to interact with a broad range of individuals and exert leadership in social interactions [67,68]. Previous studies have shown that extraversion is associated with the use of problem-solving strategies and with social support seeking [69–71].

Extroverts are prone to be achievement-oriented and goal-directed [72,73], and this may make them focus more on future time and opportunities. Moreover, extroverts are generally enthusiastic and active in social interactions [65,66]. Zimbard and Boyd (2015) [16] showed enthusiasm to be directly correlated with FTP. Extroverts actively mobilize social resources to achieve future goals [69], meaning that people with a high level of extroversion tend to be more future-oriented and have positive expectations. Accordingly, we proposed that extraversion would be positively related to FTP and have an indirect effect on career decisions via FTP, leading to these additional hypotheses:

**Hypothesis 2a.** *FTP will mediate the relationship between extraversion and career decision-making self-efficacy.*

**Hypothesis 2b.** *FTP will mediate the relationship between extraversion and career indecision.*

Conscientiousness is defined as the tendency to exhibit dependability, self-discipline, and persistence [74,75]. Individuals with high levels of conscientiousness are prudent, planned, and worthy of trust. Lee, Kelly, and Edwards (2006) [76] found conscientiousness to be negatively correlated with procrastination; conscientious people think that what they are now doing is related to the things they will do later and will lead to more opportunities. Moreover, some studies [77,78] have shown that conscientious individuals have more self-control and are better able to resist impulses, while individuals with less conscientiousness are more indifferent to the pursuit of goals, more careless, and

more hedonistic. Accordingly, we expected that conscientiousness would be positively correlated to FTP, since conscientious individuals are planners, exhibit self-control, and strive for achievement.

Furthermore, conscientiousness has been associated with several positive behavioral outcomes such as substance use, diet, and exercise, suggesting that individuals with high conscientiousness can better forego immediate pleasurable activities to focus on greater future rewards [79]. In a sample of college students, Cate and John (2007) [12] found that conscientiousness was positively associated with "the rest of opportunity". Individuals with high conscientiousness have more detailed plans for the future and react more quickly to new opportunities; they are therefore more likely to perceive of more opportunities [80,81]. Moreover, Zimbardo and Boyd (2015) [16] found a positive relationship between conscientiousness and FTP. Thus, we proposed the following additional hypotheses:

**Hypothesis 2c.** *FTP will mediate the relationship between conscientiousness and career decision-making self-efficacy.*

**Hypothesis 2d.** *FTP will mediate the relationship between conscientiousness and career indecision.*

Neuroticism is characterized by anxiety, hostility, depression, and poor inhibition of impulses [82]. Individuals with high neuroticism often experience negative emotions and lack emotional management in stressful situations [83,84]. They are more likely to be influenced by negative events and to become addicted [85]. Neurotic individuals are prone to report their health and conditions as poor [86,87]. Additionally, neurotic individuals lack the personal resources of optimism, self-efficacy, and self-esteem [69,88], which may negatively influence their evaluations of the future.

As individuals with high neuroticism are sensitive to negative events and often experience more negative emotions [83,85], their evaluations of the future may be poor. Rammstedt (2007) [89] showed that people with high neuroticism tend to think of their future as limited. In contrast, people with low neuroticism are more likely to perceive themselves and the future more positively and thus think there are many opportunities available to them Moreover, Dunkel and Weber (2010) [90] found neuroticism to be negatively related to FTP and that neurotic individuals were less satisfied with their lives. Prenda and Lachman (2001) [91] found that neuroticism had a negative effect on planning, and those with high neuroticism had fewer plans. Accordingly, we proposed that neuroticism would have an indirect effect on career decision-making self-efficacy and career indecision through FTP.

**Hypothesis 2e.** *FTP will mediate the relationship between neuroticism and career decision-making self-efficacy.*

**Hypothesis 2f.** *FTP will mediate the relationship between neuroticism and career indecision.*

To examine our hypotheses, we designed two studies. We describe these two studies separately in the following sections.

## 3. Study 1 (S1). The Relationship among Emotional and Cognitive Personality Factors, FTP and Career Decision-Making

In Study 1, we explored the mediating role of FTP on the relationship between emotional and cognitive personality traits of emotional intelligence, ego resilience, and self-control with career decision-making self-efficacy and career indecision among undergraduates. Ego resilience and self-control are cognitive factors; emotional intelligence is an emotional factor that can explain personality [37].

### 3.1. S1 Method

#### 3.1.1. S1 Participants and Procedures

For Study 1, we recruited 250 undergraduates who were studying psychology at a four-year university in Korea. The researchers explained the purpose and procedures of the research to the students, and all the students agreed to participate. Most of them carried smartphones or tablet PCs, and all participants used their own smartphones or computers to complete our online survey (posted on surveymonkey.com). The average duration for completion of the questionnaire was 10 minutes. Of the 250 participants, 125 were male (50.6%) and 122 were female (49.4%); three participants did not report their gender. In terms of college experience, 113 (45.4%) were in their first year, 82 (32.9%) in their second year, 30 (12.0%) in their third year, and 24 (9.6%) in their fourth year; one student did not report on the number of years attended. The average age was 21.58 years old (*SD* = 2.10).

The current study was conducted following Declaration of Helsinki guidelines and was approved by the Korea University review committee. We received written informed consent from all participants, in accordance with Declaration of Helsinki guidelines.

#### 3.1.2. S1 Measurements

Emotional intelligence. We measured emotional intelligence using a 16-item Emotional Intelligence Scale developed by Wong and Law (2002) [92]. The scale consists of four subscales with four items in each subscale, including a self-emotion appraisal, others' emotion appraisal, and appraisals of regulation of emotion and use of emotion. Han (2004) [93] translated the scale into Korean and adapted it in his study. Sample items included the statements "I have a good sense of why I have certain feelings," "I always know my friends' emotions from their behavior," "I am able to control my temper and handle difficulties rationally," and "I would always encourage myself to try my best"; participants rated these statements on a 5-point Likert-type scale ranging from 1 (strongly disagree) to 5 (strongly agree). In Han's (2004) [93] study, the Cronbach's alpha for this scale was 0.85. In our study, the Cronbach's alpha for the scale was 0.83.

Ego resilience. We measured ego resilience with a 14-item scale developed by Block and Kremen (1996) [94]. Ryu, Hong, and Choi (2004) [95] translated this scale into Korean, and the results from their study showed that the Korean version had good reliability. Sample items included the statements "Even when I encounter an astonishing situation, I can be all right soon and overcome it well," "I am a person full of energy," and "I am more curious than other people"; participants rated these statements on a 4-point Likert-type scale ranging from 1 (not at all) to 4 (very much). The Cronbach's alpha for this scale in Ryu et al.'s (2004) [95] study was 0.76. In our study, the Cronbach's alpha was 0.66.

Self-control. We measured self-control using a two-item scale developed by Weinstein, Healy, and Ender (2002) [96]. Sample items included the statements "I can control myself very well" and "I cannot control myself at all (reverse coded)"; participants rated these statements on a 4-point Likert-type scale ranging from 1 (not at all) to 4 (very much). The scale was translated from English to Korean following the back-translation procedure recommended by Brislin (1986) [97]. Previous studies have measured sense of self-control with a few items, as in the study of Weinstein et al. (2002) [96]; Long (1998) [98] measured the sense of self-control through a single item. In this study, the Cronbach's alpha for the scale was 0.56.

FTP. We measured the participants' FTP with the FTP Scale developed by Carstensen and Lang (1996) [99], which contains ten items. Kim (2008) [100] translated the scale from English into Korean, and the results from his study showed that the Korean version had good reliability. Sample items included the statements "There are many opportunities waiting for me in the future" and "As I get older, I begin to experience that time is limited"; participants rated the statements on a 7-point Likert-type scale ranging from 1 (not at all) to 7 (to a very great extent). The Cronbach's alpha for this scale in Park et al.'s (2018) [11] study was 0.80. In our study, the Cronbach's alpha was 0.77.

Career decision-making self-efficacy. We measured career decision-making self-efficacy with a 25-item Career Decision-Making Self-Efficacy Scale (CDMSE) developed by Betz et al. (1996) [18]. Lee (2001) [101] translated the CDMSE into Korean, and previous research has showed that the Korean version has good reliability [2,5]. The CDMSE consists of five subscales, including self-appraisal, goal selection, occupational information, planning, and problem-solving. Sample items were the statements "I can find out information related to the occupations that I am interested in via the library and the internet" and "I can choose the major that I like among the list of several majors"; participants rated the statements on a 5-point Likert-type scale ranging from 1 (not at all) to 5 (very much). In Lee's (2001) [101] study, the Cronbach's alpha for this scale was 0.85. In our study, the Cronbach's alpha for the scale was 0.66.

Career indecision. We measured career indecision with a subscale of the Career Decision Scale developed and validated by Osipow, Carney, and Barak (1976) [102]. In Korea, Ko (1992) [103] used and adapted the 18-item career indecision subscale in her study, and the Korean version has consistently shown good reliability [11,27]. A sample item was the statement "There are several professions I am equally interested in and I am having a really hard time choosing among them"; participants rated the statements on a 4-point Likert-type scale ranging from 1 (not at all) to 4 (very much).The Cronbach's alpha for this scale in Ko's (1992) [103] study was 0.86. In our study, the Cronbach's alpha was 0.69.

### 3.1.3. S1 Convergent Validity

Construct reliability was calculated to examine if the results supported the distinction of the instruments included in the research [104]. The results showed that the construct reliability for emotional intelligence was 0.81; for ego resilience was 0.78; for FTP was 0.72; for career decision-making self-efficacy was 0.78; and for career indecision was 0.87. These values suggested that their convergent validity was acceptable since the construct reliability for each measurement was greater than 0.70 [105].

## 4. S1 Analysis of Data

This study collected data based on self-reports, which may create concerns about common method bias [106]. To test for common method bias, we performed Harman's single factor test using SPSS 20.0 analysis software [107]. Since the first factor of unrotated solution explained 16.3 percent of the variance in our data, common method bias may be not particularly threatening in this study.

In this study, we used structural equation modeling with maximum likelihood estimation to test the relationships among the study variables using the AMOS™ 20 analysis software [108]. In the path structural equation model, we examined whether emotional intelligence influenced career decision-making self-efficacy and/or career indecision. We also examined whether emotional intelligence, ego resilience, and self-control had direct effects on career decision-making self-efficacy and/or career indecision through FTP, respectively. We used the comparative fit index (CFI), Tucker-Lewis index (TLI), and root mean square error of approximation (RMSEA) to assess model fit. The values of CFI and TLI > 0.90, and RMSEA < 0.08 indicated an acceptable model fit [109,110].

We performed mediation analyses using the bias-corrected bootstrapping method recommended by Preacher and Hayes (2008) [111]. When testing the mediation effect, bootstrapping was conducted with 2000 iterations and the bias-corrected confidence interval was set at 95%. If the 95% confidence interval does not include 0, then the mediation effect is considered statistically significant at the level of $\alpha = 0.05$ [112].

## 5. S1 Results

### 5.1. S1 Preliminary Analysis

Table 1 shows the means and standard deviations for the variables measured and the correlations among the variables in the study. The hypotheses in Study 1 were examined using structural equation modeling (AMOS™ 20) with the maximum likelihood estimation method. The fit indices for the

proposed model were $\chi^2$ (5, N = 250) = 13.75, CFI = 0.94, and TLI = 0.98, RMSEA = 0.08, which indicated as an acceptable fit [110].

**Table 1.** Means, standard deviations, and correlations among variables (*N* = 250).

| Variables | *Mean* | *SD* | 1 | 2 | 3 | 4 | 5 | 6 | 7 |
|---|---|---|---|---|---|---|---|---|---|
| 1. Age | 21.58 | 1.97 | – | | | | | | |
| 2. Gender [a] | 0.49 | 0.50 | −0.21 ** | | | | | | |
| 3. EI | 3.51 | 0.46 | 0.19 ** | −0.14 * | | | | | |
| 4. Ego resilience | 2.83 | 0.39 | 0.24 ** | −0.18 ** | 0.54 ** | | | | |
| 5. Self-control | 2.96 | 0.64 | 0.10 | −0.25 ** | 0.42 ** | 0.28 ** | | | |
| 6. FTP | 4.90 | 0.83 | 0.18 ** | 0.00 | 0.40 ** | 0.41 ** | 0.29 ** | | |
| 7. CDMSE | 3.60 | 0.50 | 0.26 ** | −0.07 | 0.51 ** | 0.37 ** | 0.35 ** | 0.48 ** | |
| 8. CI | 2.33 | 0.51 | −0.25 ** | −0.05 | −0.32 ** | −0.20 ** | −0.26 ** | −0.45 ** | −0.57 ** |

* $p < 0.05$, ** $p < 0.01$; [a] 1, male; 2, female; Emotional intelligence = EI; Future time perspective = FTP; Career decision-making self-efficacy = CDMSE; Career indecision = CI.

We performed path analyses based on the hypotheses (Figure 1). The results showed that emotional intelligence was positively related to FTP ($\beta = 0.20$, $p < 0.001$) and career decision-making self-efficacy ($\beta = 0.32$, $p < 0.001$), and was negatively related to career indecision ($\beta = −0.16$, $p < 0.01$). Ego resilience was positively associated with FTP ($\beta = 0.26$, $p < 0.001$). Self-control was positively related to FTP ($\beta = 0.14$, $p < 0.05$). The results also showed that FTP was positively related to career decision-making self-efficacy ($\beta = 0.18$, $p < 0.01$) and negatively related to career indecision ($\beta = −0.39$, $p < 0.001$). Finally, career decision-making self-efficacy was related negatively to career indecision ($\beta = −0.46$, $p < 0.001$).

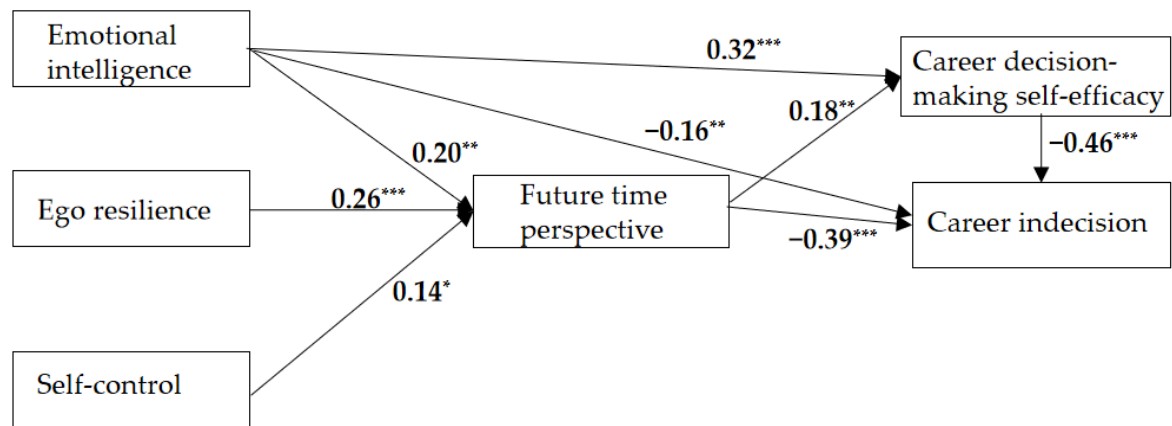

**Figure 1.** Standardized path coefficients of the model. *Note.* * $p < 0.05$, ** $p < 0.01$, *** $p < 0.001$.

### 5.2. S1 Indirect Effects

To examine the mediating effects of FTP, we followed the analysis procedure recommended by Preacher and Hayes (2008) [111]. As shown in Table 2, the indirect effect of emotional intelligence on career decision-making self-efficacy via FTP (EI→FTP→CDMSE) was statistically significant, as the 95% confidence interval from the lower to the upper bound did not include 0. We also found that the indirect effect of emotional intelligence on career indecision through FTP (EI→FTP→CI) was statistically significant, since the 95% confidence interval from the lower to the upper bound did not include 0. Therefore, emotional intelligence had direct effect on career decision-making self-efficacy and career indecision via FTP, supporting Hypothesis 1a and Hypothesis 1b. To test whether the mediating effect was partial or full, we examined the direct effects of emotional intelligence on career decision-making self-efficacy and career indecision. When the mediator was included, the predictor could significantly predict the dependent variables, which means that there was a partial mediating effect and there was a full mediating effect in the reverse [113]. In this study, emotional intelligence predicted career

decision-making self-efficacy ($\beta = 0.32$, $p < 0.001$) and career indecision ($\beta = -0.16$, $p < 0.01$), suggesting that FTP partially mediated the effect of emotional intelligence on career decision-making self-efficacy and career indecision.

In Hypotheses 1c and 1d, we proposed that FTP would mediate the effect of ego resilience on career decision-making self-efficacy and career indecision, respectively. The results showed that the indirect effect of ego resilience on career decision-making self-efficacy via FTP was statistically significant, as the 95% confidence interval of "ER→FTP→CDMSE" did not include 0. The results also showed that the indirect effect of ego resilience on career indecision through FTP (ER→FTP→CI) was statistically significant. Therefore, Hypothesis 1c and Hypothesis 1d were supported.

Hypotheses 1e and 1f proposed that self-control would have an indirect effect on career decision-making self-efficacy and career indecision via FTP. The results showed that the 95% confidence interval of "SC→FTP→CDMSE" and "SC→FTP→CI" did not include 0. Thus, FTP mediated the effect of self-control on career decision-making self-efficacy and career indecision, supporting Hypotheses 1e and Hypothesis 1f.

**Table 2.** Estimates and confidence intervals for the indirect effects among study variables.

| Direct Effect | | | Indirect Effect | | | |
|---|---|---|---|---|---|---|
| | $\beta$ | S.E | | Point Estimate | 95% CI | |
| | | | | | Lower | Upper |
| EI→CDMSE | 0.32 *** | 0.06 | EI→FTP→CDMSE | 0.14 | 0.07 | 0.23 |
| EI→CI | −0.16 ** | 0.07 | EI→FTP→CI | −0.09 | −0.16 | −0.02 |
| | | | ER→FTP→CDMSE | 0.11 | 0.04 | 0.14 |
| | | | ER→FTP→CI | −0.13 | −0.16 | −0.05 |
| | | | SC→FTP→CDMSE | 0.04 | 0.01 | 0.10 |
| | | | SC→FTP→CI | −0.04 | −0.11 | −0.01 |

*Note.* ** $p < 0.01$, *** $p < 0.001$; Emotional intelligence = EI; Ego resilience= ER; Self-control = SC; Future time perspective = FTP; Career decision-making self-efficacy = CDMSE; Career indecision = CI.

## 6. Study 2 (S2). The Mediation Effect of FTP Between Personality Traits and Career Decision-Making

In Study 2, we investigated the mediating effects of FTP on the personality traits of extraversion, conscientiousness, and neuroticism and their relationship with career decision-making self-efficacy and career indecision. While Study1 examined personality variables that could change across situations and over time, Study 2 examined stable personality traits. Hartman and Betz (2007) [114] investigated the effects of five personality factors on career decision-making self-efficacy and found a positive correlation between neuroticism and career indecision; further, extraversion and conscientiousness were demonstrated as predictors of career decision-making self-efficacy. Based on previous studies, we investigated the indirect effects of extraversion, conscientiousness, and neuroticism on career decision-making self-efficacy and career indecision via FTP.

### 6.1. S2 Method

6.1.1. S2 Participants and Procedure

In Study 2, we recruited 249 undergraduates studying psychology at a four-year university in Korea. As in Study 1, we explained the research purpose and procedures before inviting them to participate. This study was approved by the Korea University review committee. All students agreed to participate and gave their written informed consent, in accordance with to the Helsinki Declaration guidelines. The participants used their own mobile phones or computers to complete our online survey questionnaire posted at surveymonkey.com. The average duration for completion of the survey was 10 min. Of the participants, 172 were female (69.1%), 75 were male (30.1%), and two participants

did not report their gender. In terms of school experience, there were 42 (16.9%) undergraduates in their first year, 85 (34.1%) in their second year, 75 (30.1%) in their third year, and 47 (18.9%) in their fourth year; one student did not report on college experience. The average age of the participants was 21.96 years old (*SD* = 3.66).

### 6.1.2. S2 Measures

Personality traits of extraversion, conscientiousness, and neuroticism. We measured these personality traits with 30 items from the International Personality Item Pool developed by Goldberg (1999) [115]. This scale was validated in Korean by Ryu et al. (2006) [116], and their results showed high reliability. Extraversion, conscientiousness, and neuroticism were measured with ten items for each. Sample items included the statements "I am not very outstanding in a gathering (extraversion)," "If it is getting dirty, I will do cleaning immediately (conscientiousness)," and "I am not depressed at all (neuroticism)"; participants rated the statements on a five-point Likert-type scale ranging from 1 (strongly disagree) to 5 (strongly agree). In Ryu et al.'s (2006) [116] study, the Cronbach's alpha for extraversion, conscientiousness, and neuroticism were 0.69, 0.62, and 0.81., respectively. In our study, the Cronbach's alpha for extraversion, conscientiousness, and neuroticism were 0.46, 0.61, and 0.70, respectively.

FTP. As in Study 1, a 10-item scale was used to assess FTP. In Study 2, our Cronbach's alpha was.87.

Career decision-making self-efficacy. As in Study 1, we used a 25-item CDMSE to measure career decision-making self-efficacy. In Study 2, our Cronbach's alpha was 0.90.

Career indecision. We measured career indecision with the 18-item career indecision scale used in Study 1. In Study 2, our Cronbach's alpha was 0.84.

### 6.1.3. S2 Analysis of Data

We used structural equation modeling with a maximum likelihood estimation to examine the fit of the hypothesized model and the relationships among the study variables [108]. Based on the findings of Hartman and Betz (2007) [114] on the path structural equation model, we examined whether extraversion and conscientiousness influenced career decision-making self-efficacy and whether neuroticism influenced career indecision. Furthermore, we also examined whether extraversion, conscientiousness, neuroticism had direct effects on career decision-making self-efficacy and/or career indecision through FTP, respectively.

To test the mediating role of FTP, as in Study 1, we used the bias-corrected bootstrapping procedure [111]. Bootstrapping was conducted with 2000 iterations and the bias-corrected confidence interval was set at 95%; the mediating effect would be statistically significant if the 95% confidence interval did not include 0 in the level of $\alpha = 0.05$ [112].

Additionally, we conducted Harman's single-factor test to examine if common method bias was a major concern in this study [107]. The results showed that the first factor explained 18.4% percent of the variance in the data, we thus concluded that our data may be not subject to common method bias.

## 7. S2 Results

### 7.1. S2 Preliminary Analysis

Table 3 shows the means and standard deviations for the variables and the correlations among the factors in the study. The hypotheses in Study 2 were examined using structural equation modeling (Amos™ 20) with a maximum likelihood estimation method. The fit indices for the proposed model were $\chi^2$ (3, N = 249) = 6.89, CFI = 0.98, TLI = 0.92, and RMSEA = 0.07, which indicated a good fit [110].

**Table 3.** Means, standard deviations, and correlations among variables (*N* = 249).

| Variables | Mean | SD | 1 | 2 | 3 | 4 | 5 | 6 | 7 |
|---|---|---|---|---|---|---|---|---|---|
| 1. Age | 21.00 | 3.66 | | | | | | | |
| 2. Gender [a] | 0.70 | 0.46 | −0.09 | | | | | | |
| 3. Neuroticism | 3.20 | 0.75 | 0.04 | 0.13 * | | | | | |
| 4. Extraversion | 3.44 | 0.45 | −0.09 | −0.08 | −0.12 | | | | |
| 5. Conscientiousness | 3.00 | 0.46 | −0.12 | 0.08 | −0.03 | 0.22 | | | |
| 6. FTP | 4.99 | 0.81 | −0.02 | −0.08 | −0.23 ** | 0.30 ** | 0.19 ** | | |
| 7. CDSE | 3.31 | 0.47 | −0.04 | −0.09 | −0.09 | 0.27 ** | 0.27 ** | 0.52 ** | |
| 8. CI | 2.33 | 0.43 | 0.02 | 0.10 | 0.19 ** | −0.18 ** | −0.20 ** | −0.44 ** | −0.56 ** |

* $p < 0.05$, ** $p < 0.01$; [a] 1, male; 2, female; Future time perspective = FTP; Career decision-making self-efficacy = CDMSE; Career indecision = CI

We performed path analyses based on the proposed hypotheses (Figure 2). The results showed that conscientiousness was positively related to career decision-making self-efficacy ($\beta = 0.12$, $p < 0.05$). FTP was positively related to extraversion ($\beta = 0.24$, $p < 0.001$) and conscientiousness ($\beta = 0.13$, $p < 0.01$), but it was negatively related to neuroticism ($\beta = -0.20$, $p < 0.001$). Furthermore, FTP was positively related to career decision-making self-efficacy ($\beta = 0.30$, $p < 0.001$), but it was negatively related to career indecision ($\beta = -0.42$, $p < 0.001$). Finally, career decision-making self-efficacy was negatively related to career indecision ($\beta = -0.39$, $< 0.001$).

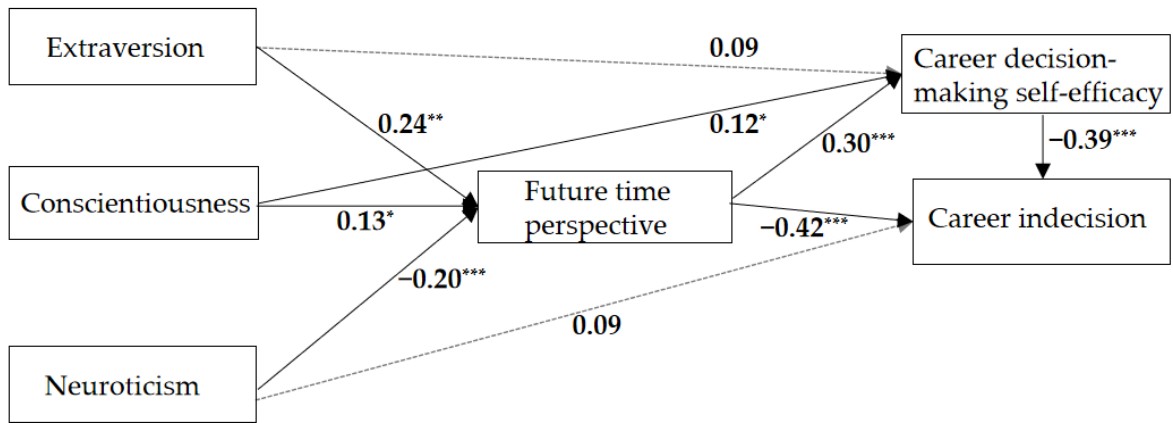

**Figure 2.** Standardized path coefficients of the model. *Note.* * $p < 0.05$, ** $p < 0.01$, *** $p < 0.001$.

### 7.2. S2 Indirect Effects

We performed bootstrapping with 2000 iterations to examine the mediating role of FTP in our data [111]. As shown in Table 4, the results showed that the indirect effect of extraversion on career decision-making self-efficacy via FTP (EXT→FTP→CDMSE) was statistically significant, because the 95% confidence interval from the lower to the upper bound did not include 0. We also found that the indirect effect of extraversion on career indecision through FTP (EXT→FTP→CI) was statistically significant, since the 95% confidence interval from the lower to the upper bound did not include 0. Therefore, extraversion had a direct effect on career decision-making self-efficacy and career indecision via FTP, supporting Hypothesis 2a and Hypothesis 2b. Moreover, the direct effect of extraversion on career decision-making self-efficacy was not significant ($\beta = 0.09$, $p =$ n.s.). This result suggests that FTP fully mediated the effect of extraversion on career decision-making self-efficacy.

In Hypotheses 2c and 2d, we proposed that FTP would mediate the effects of conscientiousness on career decision-making self-efficacy and career indecision, respectively. According to the results, the indirect effect of conscientiousness on career decision-making self-efficacy via FTP (CON→FTP→CDMSE) was statistically significant, since the 95% confidence interval from the lower to the upper bound did not include 0. The results also showed that the indirect effect of conscientiousness on career indecision through FTP (CON→FTP→CI) was statistically significant, thus supporting

Hypothesis 2c and Hypothesis 2d. Furthermore, the direct effect of conscientiousness on career decision-making self-efficacy was significant ($\beta = 0.12$, $p < 0.05$), suggesting that FTP partially mediated the effect of extraversion on career decision-making self-efficacy.

Hypotheses 2e and 2f proposed that neuroticism would have an indirect effect on career decision-making self-efficacy and career indecision via FTP. The results showed that the 95% confidence interval of "NEU→FTP→CDMSE" and "NEU→FTP→CI" did not include 0. Therefore, FTP mediated the effects of neuroticism on career decision-making self-efficacy and career indecision, supporting Hypotheses 2e and 2f. Moreover, the direct effect of neuroticism on career indecision was not significant ($\beta = 0.09$, $p$ = n.s.), suggesting that FTP fully mediated the effect of neuroticism on career indecision.

**Table 4.** Estimates and confidence intervals for the indirect effects among study variables.

| Direct Effect | | | Indirect Effect | | | |
|---|---|---|---|---|---|---|
| | $\beta$ | S.E | | Point Estimate | 95% C.I. | |
| | | | | | Lower | Upper |
| EXT→CDMSE | 0.09 | 0.05 | EXT→FTP→CDMSE | 0.12 | 0.06 | 0.20 |
| CON→CDMSE | 0.12 * | 0.05 | EXT→FTP→CI | −0.15 | −0.17 | −0.05 |
| NEU→CI | 0.09 | 0.03 | CON→FTP→CDMSE | 0.06 | 0.01 | 0.13 |
| | | | CON→FTP→CI | −0.05 | −0.11 | −0.01 |
| | | | NEU→FTP→CDMSE | −0.13 | −0.21 | −0.05 |
| | | | NEU→FTP→CI | 0.05 | 0.02 | 0.09 |

*Note*. * $p < 0.05$; Extraversion = EXT; Conscientiousness = CON; Neuroticism = NEU; Future time perspective = FTP; Career decision-making self-efficacy = CDMSE; Career indecision = CI.

## 8. Discussion

The purpose of these two studies was to examine the mediating effects of FTP among personality variables and career decision self-efficacy and career indecision. Study 1's results showed that emotional intelligence, ego resilience, and self-control had indirect effects on career decision self-efficacy and career indecision through FTP. Study 2's results showed that extraversion, conscientiousness, and neuroticism had indirect effects on career decision self-efficacy and career indecision via FTP. Thus, we make the following points.

First, personality can be divided into cognition, emotion, characteristics, and unconsciousness [37]. For that reason, we examined emotional and cognitive personality factors as predictors in Study 1. The results showed that emotional intelligence, ego resilience, and self-control had indirect effects on career decision-making self-efficacy and career indecision through FTP, supporting Hypotheses 1a–1f. The findings uncovered the role of FTP: the higher the emotional intelligence, ego resilience, and self-control, the higher the FTP will be; therefore, individuals with high levels of FTP would make their career decisions more confidently and effectively. Therefore, we found that FTP played a positive role among emotional intelligence, ego resilience, and self-control and their relationship with career decision-making self-efficacy and career indecision.

Second, we proposed that the personality traits of extraversion, conscientiousness, and neuroticism would have indirect effects on career decision-making self-efficacy and career indecision via FTP. Study 2's results supported Hypotheses 2a–2f. We confirmed the mediating effect of FTP on the relationship between extraversion and career decisions, supporting Hypotheses 2a and 2b and indicating that people high in extraversion will have a more positive FTP; further, they may show higher career decision-making self-efficacy and lower career indecision. Additionally, the mediating effects of FTP on the conscientiousness and career decisions (Hypotheses 2c and 2d) may suggest that the higher the conscientiousness of individuals, the higher their FTP, and their career decision-making self-efficacy and career indecision would also change positively. In Study 2, we also found that FTP mediated the relationship between neuroticism and career decisions (Hypothesis 2e and 2f). Accordingly, individuals

with his neuroticism may show less positive FTPs, which would reduce their career decision-making self-efficacy and increase career indecision.

### 8.1. Theoretical Implications

This study contributes to the existing literature by first investigating the relationships between personality, FTP, and career decisions. The findings revealed that FTP mediated the effect of personalities on career decision-making self-efficacy and career indecision, and this deepens our understanding of the connection between personality and career decisions. Many studies have investigated the impact of personality variables on career decision-making self-efficacy and career indecision. For instance, Di Fabio et al. (2013) [9] found a negative association between emotional intelligence and career indecision. Hartman and Betz (2007) [114] found that extraversion and conscientiousness had a positive effect on career decision-making self-efficacy and that neuroticism was negatively related to career indecision. However, little has been known about the mechanism through which individuals' personalities influence their career decision-making self-efficacy and career indecision. This study highlighted the importance of FTP, thus extending our understanding of the linkage between personality and career decisions.

Second, few studies have explored the relationship between FTP and personality [16,23,117], so little understanding has been gained about the linkage between FTP and personality. FTP is a cognitive-motivational construct that describes individuals' sense of future time, and affects such behaviors as planning, goal setting, and processes like autobiographic memory [21,78]. Emotional intelligence (emotional factor) and the Big Five are personality traits, the units of personality [105]. FTP is more likely to be a cognitive construct. By examining the relationship among FTP and emotional and cognitive personality traits, this study extends the FTP literature and shows that the level of positivity in individuals' FTP can inform our understanding of understand their personality. Moreover, whereas previous research has mainly focused on the consequences of FTP [80,81,118], little attention has been paid to the antecedents of FTP. Our findings indicate that individuals' emotional and cognitive personality factors of emotional intelligence, ego resilience, and self-control and the personality traits of extraversion, conscientiousness, and neuroticism can influence their FTP: People with good emotional management or the ability to control themselves are expected to possess a more positive FTP.

### 8.2. Practical Implications

Our results have some important practical implications for career counseling. The findings showed that FTP mediates personality variables and career variables; personality can positively affect career variables by increasing students' FTP positivity. Therefore, schools can improve students' career decision-making and reduce their career indecision by developing and implementing programs that increase student's FTP positivity—that is, by helping them understand that positive actions, efforts, and behaviors now can lead to more positive results and brighter occupational futures. Previous studies [50,119–121] showed that programs designed to increase FTP positivity resulted in increased career decision-making self-efficacy and reduced career indecision. For example, such a program might involve counselors having students write motivational letters to themselves about achieving goals and success [5]. Furthermore, research has shown that a significant portion of undergraduates today are not future oriented, and they rarely contemplate their career path before or during college; this can lead to increased unemployment rates [1,122]. Therefore, it would be helpful for schools, society, and policymakers to encourage young adults to foster positive attitudes toward their future, explore career opportunities, and prepare for their future careers. By taking part in FTP positivity programs, job seekers nearing graduation from universities may be motivated to increase their job search efficacy and act to realize career goals, and this may contribute to more sustainable career management. Moreover, if programs designed to enhance FTP are implemented for employees who are about to retire, their anxiety over choosing a second career would be reduced, allowing them to prepare for their next careers more confidently or to expand or modify their existing careers after retirement.

*8.3. Limitation and Future Directions*

This study had some limitations. First, both Study 1 and Study 2 used a cross-sectional design to investigate the relationships among study variables, and that made it difficult to ascertain causal relationships. To determine causation among study variables, future research should measure the predictors, the mediator, and the outcomes separately by setting a certain time interval and measuring at least three times; such longitudinal study might reveal the casual relationships. Second, in Study 1, self-control was measured with the following two items: "I can control myself well" and "I cannot control myself at all." Although a previous study [96] used these items, there may be some issues in the validity and reliability of the scale. Considering that the Cronbach's alpha of this scale was 0.56 in this study, future studies should use a strict standardized scale. Third, the Cronbach's alpha of the scales used in Study 1 and the two scales used in Study 2 were lower than 0.70. Generally, to make a research scale acceptable, the Cronbach's alpha should be over 0.70 [123]. However, the scales for ego resilience, self-control, career decision-making self-efficacy, and career indecision in Study 1 and the scales for neuroticism and conscientiousness in Study 2 did not reach this standard; this limits the generalizability of our study results. To generalize the results obtained in this study, future studies need to examine whether the data collected by these scales and other scales support our theoretical model.

Finally, the FTP in this study was a construct measuring a general time perspective. Bandura (1977) [124] divided self-efficacy into general self-efficacy, task-specific self-efficacy, and situation-specific self-efficacy. The career decision-making self-efficacy in this study was task-specific. As with the divisions used for self-efficacy, it would be beneficial to divide FTP into general and situation specific. For example, supposing there is one female university student whose family owns a shop likes planning and has a good mastery of figures but does not study hard or enjoy taking risks; although her career FTP might not be very positive (she is unlikely to picture herself as CFO of a major firm), her personal FTP might still be positive (she may see that by taking over the family business, she will please her family). Similar to the necessity of field-specific FTP, Zacher and Frese (2009) [80] developed measures of occupational FTP and conducted validation studies. This scale was adapted for employees who had jobs, but it cannot be used for students or people searching for jobs. Accordingly, future studies should develop and validate the career FTP scale and conduct study among diverse samples.

## 9. Conclusions

The present study contributes to knowledge about the associations between personality factors and career decisions by investigating the mediating role of FTP. Among our participants, emotional personality (i.e., emotional intelligence), the cognitive personality traits of ego resilience and self-control, and the emotional personality traits of extraversion, conscientiousness, and neuroticism influenced career decision-making self-efficacy and career indecision through FTP. This study furthers understanding of the antecedents and consequences of FTP and provides implications for sustainable career management.

**Author Contributions:** I.-J.P. developed research idea/hypotheses, conducted initial data analysis, and wrote the first draft; M.G. wrote the second draft and responded the first revision. S.H. supervised M.G. and responded the first and second revision. All authors have read and agreed to the published version of the manuscript.

**Funding:** This research received no external funding.

**Conflicts of Interest:** The authors declare no conflict of interest.

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
