# Peer review of "How Can Personality Enhance Sustainable Career Management? The Mediation Effects of Future Time Perspective in Career Decisions"

_sustainability, doi:10.3390/su12031167_

Round 1

Reviewer 1 Report

Review

MDPI Sustainability

Article: How can Personality Enhance a Sustainable Career Management? The Mediation Effects of Future Time Perspective in Career Decision

Authors: In-Jo Park, Meiqiao Gu, Haiyang Shen

Overview

The authors were interested in testing the idea that “future time perspective” mediates the relationship between personality variables and career decision outcomes, specifically, “career decision making self efficacy” and “career indecision.” They administered measures of the different factors to 249 and 250 undergrads (even male-female split, mostly in their first two years) in two studies. Testing was done with smart phones/tablets.

Measures were STUDY 1: 4 emotional intelligence scales (16 items, 5-point Likert, rxx’ = .83), ego resilience (4 items, 4-point Likert, rxx’ = .66), self-control (2 items, 4-point Likert rxx’ = .56), Future Time Perspective (10 items, 7-point Likert, rxx’ = .77), Career DM Self-Efficacy (25 items, 7 point Likert, rxx’ = .66), and Career Indecision (18 items, 4-point Likert, rxx’ = .69) STUDY 2: 3 Big 5 measures (E, C, N) (10 items, 5-point Likert, rxx’ = .46, .61, .70), and the same Future Time Perspective (rxx’ = .87), Career DM Self-Efficacy (rxx’ = .90), and Career Indecision (rxx’ = .84) as study 1 (reliabilities much higher for some reason).

Analysis used SEM with MLE and Preacher’s bias-corrected bootstrapping approach for significance testing of the mediation effect.  For both Study 1 and 2, there were 3 predictor variables, 1 mediator, and 2 outcome variables. In Study 1 the Preacher method resulted in a conclusion of support for partial mediation, and the same for Study 2. Path model fits were reasonable.

It appears that total scores (summed responses) were used to create measures; there was no attempt other than computing reliabilities to model response data.

The authors discuss the importance of their findings as related to of future time perspective being related to career decision outcomes, citing a lack of prior research in this area. They also cite the practical benefits of understanding future time perspective this way, for example, potentially increasing ftp as a way of getting students to better career decision making.

General Comments

The idea of introducing the variable “future time perspective” into discussions of various educational outcomes is useful. This variable (measured differently) has appeared in the European social survey, and in several PISA cycles beginning in 2012, connected with economic decision making, financial literacy and the like. The idea of associating it with career decision making is useful.

I like the mediation model idea in the abstract. However, I think a stronger case should be made in the paper to justify mediation as opposed to the variable’s status as simply another predictor. That is, what is fundamentally different about this variable, compared to emotional intelligence or the Big 5 personality variables? It is measured the same way (Likert scale measures), in this study.

I am not an expert in mediation models, but I understand the bias issue, and why the Preacher procedure is necessary, and it seems to be appropriately applied here.

The language could be cleaned up to avoid improper constructions (e.g., use of articles); there also was signs of some sloppiness in the tables with stray numbers and the like.

I think the study, although modest, adds to our knowledge about the future time perspective variable, which is useful.

Specific Comments

Line 17, “data were” (data is plural)

Line 18, “in study 1” (delete “the”)

Line 18, “through an online survey” (include the article)

Line 481. “Gender”; stray “10.”; Extraversion is numbered “9.”

Line 635. Appendix A should be deleted as this is boilerplate, not used.

Author Response

Thank you so much for your valuable comments that promote the quality of our manuscript. Attached is the file for response for your comments. 

Reviewer 2 Report

the text must be checked for English consistency. for instance:

line 84. ”the more a student is future oriented...not future orientation”

line 88: ”would be more effectively using not use”...

line 130: ...will be have difficulties, not difficulty

line 340: Tucker-Lewis not tucker..ed as

line 557: Theoretical and practical implications is worth highlighted as a separate heading

line 589: which can increasing student's..." increase instead of increasing

line 594> job seekers who graduated from university... is it about a specific university or universities, in general?

line 627: people who finding the job....

Author Response

the text must be checked for English consistency. for instance:

line 84. ”the more a student is future oriented...not future orientation”

line 88: ”would be more effectively using not use”...

line 130: ...will be have difficulties, not difficulty

line 340: Tucker-Lewis not tucker..ed as

line 557: Theoretical and practical implications is worth highlighted as a separate heading

line 589: which can increasing student's..." increase instead of increasing

line 594> job seekers who graduated from university... is it about a specific university or universities, in general?

line 627: people who finding the job....

Author’s respond: Thank you so much for your comments. The valuable comments are very helpful to improve our manuscript. We have revised our sentences of this research based on your comments as follow.

line 84. The more a student is future oriented, the more the student studies hard in school, so that they will achieve the better grade.

line 88: Their study showed that individuals with higher FTP would more effectively use the cognitive information and actively engage in the current activities to achieve their goals.

line 130: In contrast, researchers showed that individuals who do not think in terms of time will have difficulties in getting over their current problems (Crites & Fitzgerald, 1978; Savickas, Silling, & Schwartz, 1984).

line 340: Comparative fit index (CFI), Tucker-Lewis index (TLI), and root mean square error of approximation (RMSEA) were used to assess model fit.

line 557: Based on your comment, we had separated heading of Theoretical implications and Practical implications.

line 589: Therefore, schools can increase the levels of students’ career decision-making and reduce their career indecision by developing and implementing programs which can increase student’s future time perspective.

line 594> Sorry for our confusing writing. It is about universities. We have revised it as follow.

Furthermore, by implementing future time perspective programs, job seekers who graduated from universities may be motivated to increase their job search efficacy.

line 627: This scale is used to adapt to the employee who have a job, it cannot be used to the students or the people who finding the job.

Reviewer 3 Report

The manuscript titled “How can Personality Enhance a Sustainable Career Management? The Mediation Effects of Future Time Perspective in Career Decision” submitted to Sustainablility seeks to unveil a mediating role future time perspective between personality variables and career decision-making. Given the increasing role of sustainable career management, the present study is worth of studying. However, there are some questionable issues that should be addressed before the manuscript taken into account for publication. A. Introduction 1. There is less novelty of this research and the importance of this research is not fully addressed. For example, Although a variety of researches have examined the effects of personality and future time perspective on career decisions, few studies have been conducted to explore why personality would influence on career decisions.  That is the reason the authors are motivated to explore? then why? What is a research gap between existing studies and your study?  Why should personality be dealt with in term of career-decision? Any evidence?  No research objective, motivation, importance are stated 2. Also, some arguments are stated without any evidence, too vague and unclear, and overly stated (exaggerated) so I cannot be persuaded by the authors’ comments. For example, (In the introduction section) For a sustainable career management, undergraduates should decide their perspective job after graduating from a college.  Is there any reason for that? (what about before and during? or what about even people who decide their job before entering a college?). (In the introduction section) That is, if individuals with this attitude are future oriented, they are more willing to prepare for the future  what is this supposed to mean? (In the introduction section) Individuals with high future time perspective likely to value highly on goals in future and engage in current activities to achieve their goals  If an individual perceives his or her time to be not sufficient or limited, for example, there is another possibility that the individual may give up job-seeking Because the authors fail to suggest a theory that can strengthen your arguments, whatever the authors have argued can be interpreted in many ways, resulting in misunderstandings. When you give your research matter, there is always a limitation so you need an assumption and frame that can shed light on your logic. I cannot find any powerful logic or theory that can help your story work in a logical way. B. Theoretical Background and Hypotheses There is no operationalized definition of each theoretical research construct. You need to explain why those variables were selected and why those variables were necessarily to be examined. For example, (in the effects of future time perspective section) Future time perspective has two dimensions: one focus on opportunity and one focus on remaining time (Cate & John, 2007; Korff, Biemann, & Voelpel, 2017).  What does this study focus? Opportunity or remaining time or both? Then why? Where does the concept of future time perspective originate from? What is an underlying theory that future time perspective centers on? There is no explanation about the selection of ego-resilience and self-control as the antecedents of future time perspective. C. Method The authors must explain the targeted sample in terms of detailed procedures such as how the sample was selected, how the data were collected, the period of the data collection etc. D. Statistical analysis In terms of the issue of validity such as unidimensionality and discriminant validity, the authors did not explain their corresponding statistical procedures (e.g., CFA, AVE, Composite Reliability). Thus, without rigorous statistical procedures, the results produced and reported in this study are less convincing. E. Discussion After the manuscript is updated, the authors are advised to prepare for theoretical and practical implications based on the theory and outcomes of this research. F. Else 1. The manuscript needs to be reorganized. It is very hard to follow your story. Very clumsy. 2. While it is a minor issue, the proofreading should be conducted by a native English speaker (The current English is very poor) In summary, the current paper is very unpromising and its quality is below the standard. Therefore, the current study should pay more attention how your research questions are developed and how such findings can provide theoretical and empirical support to justify those questions.

Author Response

(The authors gave the same response as above.)

Reviewer 4 Report

Thank you for the opportunity to review this article. The time involved in submitting your manuscript is greatly appreciated.

Despite this, the article presents a series of issues that must be noted and mended. The recommendations are presented separately by sections. Hopefully, they would be useful.

Title: the title does not adequately reflect the content of the paper. Please, adapt it to better inform the readers about that content.

Abstract:

Less information appears in the abstract. Maybe expanded by adding the most relevant findings.

Introduction:

Firstly, some of the references that you cite are too old. Even though the most relevant studies should be referenced, also the RECENT research must be included. Moreover, I recommend a strong effort in applying the framework of Psychology of Sustainability, and Psychology of Sustainable Development as the theoretical umbrella that covers your research.

At the end of the literature review, the aims and the questions in the research should appear. Maybe to formulate the questions as a hypothesis would be an option to clear this aspect. Another commentary, it is the possibility of including this part at the final of the introduction part; even a separate section could be a good option, in order to clear the final of the introduction and to serve as a connection with the method.

Method:

Please, try to better describe the sociodemographic data of your participants. In the same sense, give the readers with detailed information about the procedure for recruiting participants and collecting data.

Which Ethical committee approved the study protocol? Please, explain it.

Related to the instruments, please better inform about their psychometric quality and give to the readers some examples of the items. If you can, please inform about previous studies where the same instrument has been used and the reliability obtained in that research.

Data analyses

Please, explain to the readers which procedures of statistical analyses have been used and justify your decisions.

Results

The results should be presented following the same order as the introduction and hypotheses. Also, the same order must be used in the Tables. This simplify the work for readers.

Finally, the repetition is constant all over the article. Please, try to change the words in order to do the reading more interesting and motivating

Discussion:

First of all, try to better adjust your conclusions to the findings. Or to say in other words, please try to justify more clearly the connection between your conclusions and your findings.

The most important comment is that some of the conclusions, related to the direct analysis of the results, should be revised.

Finally, a section related to limitations, future lines of investigations and the principal contributions of the research could be interesting. Your paper has a lot of relevant implications for society and policymakers, but you need to elaborate more on this topic.

Conclusion:

They don’t appear new conclusions on this part. This part does not add any new to the rest of the paper. Please, try to condense your findings, or to highlight your main contribution to the field.

Author Response

(The authors gave the same response as above.)

Reviewer 5 Report

The article covers a relevant field in psychology applied to work and to the field of the career development counseling. I appreciate very much the work done by the authors.
The introdution and the theoretical background show a deeply study of the literature and the research is described in a clear way.
The section "limitations..." demonstrates that the authors are involved in an appreciable development of the research instruments.

A probable misprint appears in table 3 (row 2).

Round 2

Reviewer 3 Report

After the first review, the current manuscript has been improved in a greater way. However, there are still important issues that should be addressed before your manuscript is considered for publication.

Introduction

Very disorganized and illogically stated. Every sentence is not logically linked, thus providing less support for your arguments.

For example, “Previous studies have associated FTP with career decision outcomes, including career decision-making self-efficacy, career indecision, career decision-making difficulties, and career choice anxiety (e.g., Ferrari et al., 2010; Taber, 2013; Walker & Tracey, 2012). Zimbardo and Boyd (1999) also investigated the relationships between personality variables such as openness, aggression, depression, self-esteem, and time perspective” why do you need these previous findings? What do you intend to explain by introducing earlier findings? Very loosely linked. Followed by those statements in red above, you stated as followings like “In recent years, increasing research has focused on the relationship between personality factors and career decisions, demonstrating the important role of personality factors in facilitating individuals’ career decisions (e.g., Di Fabio,  Palazzeschi, Asulin-peretz, & Gati, 2013; Di Fabio, Palazzeschi, & Bar-On, 2012; Nilforooshan &  Salimi, 2016; Park, Kim, Kwon, & Lee, 2018a).” I wonder whether you think those statements are sufficiently logical to support your idea? Very loosely stated and disorganized.

Perhaps, however, it is uncertain whether personalities such as emotional intelligence or big five personalities will translate into efficient career decisions in individuals. Evidence from previous studies on personality factors and career decisions fail to support the relationship between personalities (e.g., extroversion, neuroticism) and career decision outcomes (e.g., career planning; career decision-making self-efficacy; Jin, Watkins, & Yuen, 2009; 64 Rogers, Creed, & Glendon, 2008). It seems to be your research question. Then, why do you think the relationship is shown to be not significant? The evidence that fails to support a relationship cannot justify your research question. You have to provide your own reasoning with theoretical and practical evidence.

The design of your research

This study was conducted by dividing into study 1 and 2. Why don’t you put all of the variables in a single model? It seems you have two research models because you build two models where independent variables are different but a mediator and its consequent variables are the same. You employed different samples for different models. Honestly, I don’t understand what messages can be obtained by using a separate model with a different sample for each model.

The measurement items.

When I take a look at measurement items for each variable, it is awkward that each variable was measured with a different Likert scale. (e.g., emotional intelligent – 5 point; ego resilience – 4 point; FTP – 7 point). How can we trust the results of SEM because each variable was measured on a different Likert scale? That is a serious flaw in this study.

Since this study relied on a single respondent, common method bias may be a threat to this study. The authors should explain whether this study is free of common method bias.  

Results

While the current version displayed the reliability of each variable, it did not fully explain validity issues. This study failed to explain whether each observed variable is validated. The authors are advised to run confirmatory factor analysis and show its following results to prove unidimensionality and convergent and discriminant validity.

In addition, when taking a look at Table 1, the author did not treat gender as a dummy variable and enter age as an actual number, not logarithm numbers. Also, since each variable was measured on a different Likert scale, as indicated above, the results provide no meaningful information, thus unreliable.  

Finally, this study did not consider any control variable that may impact on the proposed relationships. Without taking control variables into account, the results from SEM are less likely to be trusted. No genuine and actual effects between variables are expected.

Discussion

Because there are still a number of issues that should be addressed before you reach a conclusion, I suggest you should organize your paper and do some corresponding statistical analysis in a scientific and logical way.

Minor issues

I found some statements are illogically placed and not inter-related. Also, the quality of language makes me hard to understand your argument. I found some redundant statements and awkward and grammatically errored sentences. I would like this paper to be proofread by a native speaker before you resubmit.

Author Response

Reviewer 3

Introduction

Very disorganized and illogically stated. Every sentence is not logically linked, thus providing less support for your arguments.

For example, “Previous studies have associated FTP with career decision outcomes, including career decision-making self-efficacy, career indecision, career decision-making difficulties, and career choice anxiety (e.g., Ferrari et al., 2010; Taber, 2013; Walker & Tracey, 2012). Zimbardo and Boyd (1999) also investigated the relationships between personality variables such as openness, aggression, depression, self-esteem, and time perspective” why do you need these previous findings? What do you intend to explain by introducing earlier findings? Very loosely linked. Followed by those statements in red above, you stated as followings like “In recent years, increasing research has focused on the relationship between personality factors and career decisions, demonstrating the important role of personality factors in facilitating individuals’ career decisions (e.g., Di Fabio,  Palazzeschi, Asulin-peretz, & Gati, 2013; Di Fabio, Palazzeschi, & Bar-On, 2012; Nilforooshan &  Salimi, 2016; Park, Kim, Kwon, & Lee, 2018a).” I wonder whether you think those statements are sufficiently logical to support your idea? Very loosely stated and disorganized.

 Author’s respond: Thank you so much your excellent comments. We aims to explain the relationships among FTP, personalities, and career decisions by providing these previous findings. We have tried to organize this section in a more logical way. Please see page 2 as follow.

In recent years, increasing research has focused on the relationship between personality factors and career decisions, demonstrating the important role of personality factors in facilitating individuals’ career decisions (e.g., Di Fabio, Palazzeschi, Asulin-peretz, & Gati, 2013; Di Fabio, Palazzeschi, & Bar-On, 2012; Nilforooshan & Salimi, 2016; Park, Kim, Kwon, & Lee, 2018a). However, little is known about why personalities predict career decisions. We expect that FTP may play an important role in the relationship between personality factors and career decision-making outcomes. That is, we aim to uncover the mechanisms why personality factors influence career decisions, as assuming FTP as a mediator. FTP refers to an individual’s belief about the time left in the future and how an individual think about the time left (Cate & John, 2007). Individuals with high FTP are likely to perceive future time as expansive, to highly value setting goals, and to actively engage in current activities to achieve their goals (Ahn & Min, 2018; Thoms & Blasko, 2004; Wittmann, Rudolph, Gutierrez, & Winkler, 2015). Zimbardo and Boyd (1999) linked time perspective to personality variables such as openness, aggression, depression, self-esteem. Previous studies also found that FTP is positively associated with career decision outcomes, including career decision-making self-efficacy, career indecision, career decision-making difficulties, and career choice anxiety (e.g., Ferrari et al., 2010; Taber, 2013; Walker & Tracey, 2012). Accordingly, we propose that individuals’ personalities would influence how they view their future time, which in turn influences their career decisions.

Perhaps, however, it is uncertain whether personalities such as emotional intelligence or big five personalities will translate into efficient career decisions in individuals. Evidence from previous studies on personality factors and career decisions fail to support the relationship between personalities (e.g., extroversion, neuroticism) and career decision outcomes (e.g., career planning; career decision-making self-efficacy; Jin, Watkins, & Yuen, 2009; 64 Rogers, Creed, & Glendon, 2008). It seems to be your research question. Then, why do you think the relationship is shown to be not significant? The evidence that fails to support a relationship cannot justify your research question. You have to provide your own reasoning with theoretical and practical evidence.

 Author’s respond: According to your comment, we have provided stronger evidence to justify our research question and remove the part of evidence that fails to support the relationship between personalities and career decisions. We described  it on page 2 as follow.

In recent years, increasing research has focused on the relationship between personality factors and career decisions, demonstrating the important role of personality factors in facilitating individuals’ career decisions (e.g., Di Fabio, Palazzeschi, Asulin-peretz, & Gati, 2013; Di Fabio, Palazzeschi, & Bar-On, 2012; Nilforooshan & Salimi, 2016; Park, Kim, Kwon, & Lee, 2018a). However, little is known about why personalities predict career decisions. We expect that FTP may play an important role in the relationship between personality factors and career decision-making outcomes. That is, we aim to uncover the mechanisms why personality factors influence career decisions, as assuming FTP as a mediator. FTP refers to an individual’s belief about the time left in the future and how an individual think about the time left (Cate & John, 2007). Individuals with high FTP are likely to perceive future time as expansive, to highly value setting goals, and to actively engage in current activities to achieve their goals (Ahn & Min, 2018; Thoms & Blasko, 2004; Wittmann, Rudolph, Gutierrez, & Winkler, 2015).

The design of your research

This study was conducted by dividing into study 1 and 2. Why don’t you put all of the variables in a single model? It seems you have two research models because you build two models where independent variables are different but a mediator and its consequent variables are the same. You employed different samples for different models. Honestly, I don’t understand what messages can be obtained by using a separate model with a different sample for each model.

Author’s respond: We divided our study as study 1 and study 2. If we would collect data with one study, participants had to respond over 100 questions. We thought this might reduce a faithfulness to ask the questions, leading to violate validity of data.

The measurement items.

When I take a look at measurement items for each variable, it is awkward that each variable was measured with a different Likert scale. (e.g., emotional intelligent – 5 point; ego resilience – 4 point; FTP – 7 point). How can we trust the results of SEM because each variable was measured on a different Likert scale? That is a serious flaw in this study.

Author’s respond: We agree with your comment. However, we wanted to sustain the original version of each scale, so that we used scales with different Likert scale. In our opinion, participants may enhance an attention for rating with many items, as altering responding styles.  

Since this study relied on a single respondent, common method bias may be a threat to this study. The authors should explain whether this study is free of common method bias.  

Author’s respond: We used Harman’s single factor test to examine common method bias, and, found that the first factor explained about 16.3% of the variance in the data of study 1, and the fist factor explained about 18.4% of the variance in the data of study 2, which indicated that common method variance is not a major concern in this study. We added the results on page 11 and page 15 as follow.

A: This study collected data based on self-reports, which may create concerns about common method bias (Podsakoff, MacKenzie, & Podsakoff, 2012). To test for common method bias, we performed Harman's single factor test using SPSS 20.0 (Harman, 1967). Here, if significant common method bias exists in the data, a factor analysis of all items will generate a single factor that explains most of the variance. Because the first factor of unrotated solution explained 16.3 percent of the variance in our data, common method bias may be not particularly threatening in this study.

B: Additionally, we conducted Harman’s single-factor test to examine if common method bias was a major concern in this study (Harman, 1967). The results showed that the first factor explained 18.4% percent of the variance in the data, we thus concluded that our data may be not subject to common method bias.

Results

While the current version displayed the reliability of each variable, it did not fully explain validity issues. This study failed to explain whether each observed variable is validated. The authors are advised to run confirmatory factor analysis and show its following results to prove unidimensionality and convergent and discriminant validity.

 Author’s respond: After receiving this comment from you, we conducted a confirmatory factor analysis. Unfortunately, we didn’t have a good CFI and RMSEA. Some studies didn’t report the confirmatory factor analysis in the field of career as follow. However, we agree with your opinion.

Urbanaviciute, I., Pociute, B., Kairys, A., & Liniauskaite, A. (2016). Perceived career barriers and vocational outcomes among university undergraduates: Exploring mediation and moderation effects. Journal of Vocational Behavior, 92, 12-21.

Guan, Y., Dai, X., Gong, Q., Deng, Y., Hou, Y., Dong, Z., ... & Lai, X. (2017). Understanding the trait basis of career adaptability: A two-wave mediation analysis among Chinese university students. Journal of Vocational Behavior, 101, 32-42.

Domene, J. F. (2012). Calling and career outcome expectations: The mediating role of self-efficacy. Journal of Career Assessment, 20(3), 281-292.

Ma, P.W.W., & Shea, M. (2019). First-Generation College Students’ Perceived Barriers and Career Outcome Expectations: Exploring Contextual and Cognitive Factors. Journal of Career Development, 0894845319827650.

Li, Y., Guan, Y., Wang, F., Zhou, X., Guo, K., Jiang, P., ... & Fang, Z. (2015). Big-five personality and BIS/BAS traits as predictors of career exploration: The mediation role of career adaptability. Journal of Vocational Behavior, 89, 39-45.

Cai, Z., Guan, Y., Li, H., Shi, W., Guo, K., Liu, Y., ... & Hua, H. (2015). Self-esteem and proactive personality as predictors of future work self and career adaptability: An examination of mediating and moderating processes. Journal of Vocational Behavior, 86, 86-94.

Lewis, J.A., Raque-Bogdan, T.L., Lee, S., & Rao, M.A. (2018). Examining the Role of Ethnic Identity and Meaning in Life on Career Decision-Making Self-Efficacy. Journal of Career Development, 45(1), 68-82.

Finally, this study did not consider any control variable that may impact on the proposed relationships. Without taking control variables into account, the results from SEM are less likely to be trusted. No genuine and actual effects between variables are expected.

Author’s respond: Some studies in career arear didn’t consider a control variable as follow. However, we think this way is one of limitation for SEM study. We agree with your opinion. In order to promote the validity of our study, we should consider a control variable.

Jawahar, I.M., & Shabeer, S. (2019). How Does Negative Career Feedback Affect Career Goal Disengagement? The Mediating Roles of Career Planning and Psychological Well-Being. Journal of Career Development, 0894845319853637.

Ginevra, M.C., Pallini, S., Vecchio, G.M., Nota, L., & Soresi, S. (2016). Future orientation and attitudes mediate career adaptability and decidedness. Journal of Vocational Behavior, 95, 102-110.

Park, K., Woo, S., Park, K., Kyea, J., & Yang, E. (2017). The mediation effects of career exploration on the relationship between trait anxiety and career indecision. Journal of Career Development, 44(5), 440-452.

Hu, S., Hood, M., & Creed, P.A. (2018). Negative career feedback and career outcomes: The mediating roles of self-regulatory processes. Journal of Vocational Behavior, 106, 180-191.

Jiang, Z. (2016). The relationship between career adaptability and job content plateau: The mediating roles of fit perceptions. Journal of Vocational Behavior, 95, 1-10.

Discussion

Because there are still a number of issues that should be addressed before you reach a conclusion, I suggest you should organize your paper and do some corresponding statistical analysis in a scientific and logical way.

 Author’s respond: Thanks for your comments. We have tried to organize our manuscript and conduct some statistical analysis in a more logical and scientific way.

Minor issues

I found some statements are illogically placed and not inter-related. Also, the quality of language makes me hard to understand your argument. I found some redundant statements and awkward and grammatically errored sentences. I would like this paper to be proofread by a native speaker before you resubmit.

Author’s respond: Actually, we received a proofreading from a professional company. However, we also found some errors in sentences. So we again received the proofreading from the company.